# Cellular mechano-environment regulates the mammary circadian clock

Nan Yang[1,2], Jack Williams[1,2], Vanja Pekovic-Vaughan[1], Pengbo Wang[1,2], Safiah Olabi[1,2], James McConnell[1], Nicole Gossan[1], Alun Hughes[1], Julia Cheung[1,2], Charles H. Streuli[1,2,*] & Qing-Jun Meng[1,2,*]

Circadian clocks drive ~24 h rhythms in tissue physiology. They rely on transcriptional/translational feedback loops driven by interacting networks of clock complexes. However, little is known about how cell-intrinsic circadian clocks sense and respond to their microenvironment. Here, we reveal that the breast epithelial clock is regulated by the mechano-chemical stiffness of the cellular microenvironment in primary cell culture. Moreover, the mammary clock is controlled by the periductal extracellular matrix *in vivo,* which contributes to a dampened circadian rhythm during ageing. Mechanistically, the tension sensing cell-matrix adhesion molecule, vinculin, and the Rho/ROCK pathway, which transduces signals provided by extracellular stiffness into cells, regulate the activity of the core circadian clock complex. We also show that genetic perturbation, or age-associated disruption of self-sustained clocks, compromises the self-renewal capacity of mammary epithelia. Thus, circadian clocks are mechano-sensitive, providing a potential mechanism to explain how ageing influences their amplitude and function.

---

[1] Faculty of Biology, Medicine and Health, University of Manchester, Oxford Road, Manchester M13 9PT, UK. [2] Wellcome Centre for Cell-Matrix Research, University of Manchester, Oxford Road, Manchester M13 9PT, UK. * These authors contributed equally to this work. Correspondence and requests for materials should be addressed to C.H.S. (email: cstreuli@manchester.ac.uk) or to Q.-J.M. (email: Qing-Jun.Meng@manchester.ac.uk).

Cell-autonomous circadian clocks in the brain and periphery drive ~24 h rhythms in fundamental biological processes that control tissue physiology, including metabolism, cell proliferation, differentiation, cell cycle and stem cell function[1,2]. At the molecular level, circadian oscillations rely on a transcription–translation feedback loop driven by a core clock mechanism. This clock consists of the BMAL1/CLOCK trans-activation complex, the Period (PER)/Cryptochrome (CRY) repressive complex, and the auxiliary REV-ERB/ROR-stabilising loop[1–5]. The robustness of circadian rhythms in multiple tissues deteriorates with ageing, compromising the temporal control of physiology[6–10]. Age-associated clock suppression may be a predisposing factor for various human diseases.

However, our understanding of how young cellular clocks maintain robust circadian outputs, and how this robustness is lost during ageing, remain largely unknown. Previous studies have revealed the paramount importance of maintaining a robust 24 h circadian rhythm that is synchronized with daily environmental changes[1,5]. Not surprisingly, the intricate molecular oscillator is built with the capacity to respond to multiple environmental and metabolic time cues, such as the light/dark cycle, feeding/fasting rhythm, body temperature fluctuations and daily surges of hormones.

Here, we have systemically characterized circadian clock mechanisms in mammary gland biology. Our data reveal a novel link between circadian clock genes and mammary stem cell function. Moreover, we have identified a new regulatory mechanism for the mammary epithelial clock, which occurs through the mechano-stiffness of the cellular microenvironment. In summary, our work first reveals a new function for cell-matrix interactions, which is that it regulates circadian biology. Second, it shows that tissue stiffening suppresses the mammary circadian clock activity *in vivo*, which could contribute to an increased risk for breast diseases and even cancer.

## Results

**The mammary clock controls tissue-specific rhythmic genes.**
The mammary gland consists of a branching network of epithelia, ensheathed in a basement membrane and surrounded by stroma[11]. Real-time bioluminescent imaging of mammary tissue explants from PER2::Luc clock reporter mice[12] revealed daily rhythmic variations, where for example PER2::Luc was expressed strongly at 24 h but very weakly 12 h later (Fig. 1a). This is consistent with the strong expression of core clock proteins BMAL1 and PER2 in mammary ducts (Fig. 1b).

To gain insights into the role of the mammary clock, circadian time-series microarrays were performed to identify rhythmic genes *in vivo*. Mammary tissues were isolated at 4 h intervals for two circadian (24 hourly) cycles, from mice kept under constant darkness to avoid any light- or dark-driven genes. Remarkably, 594 genes were under circadian control. A subset of clock-controlled genes included those linked to progenitor/epithelial cell function, for example, α6-integrin, Prkcε, P21 or Bcar3[13–16], whose rhythmic expression was validated by qRT-PCR (Fig. 1d). Many rhythmic genes peaked around the day/night transition (Figs 1c and 2a, Supplementary Table 1). Significant GO-term clusters include 'Metal ion binding/transcription factors', 'Nuclear Hormone Receptors', 'Src homology-3 domain', 'Positive regulation of macromolecule biosynthetic process' and 'Lysosome' (Fig. 2b). We have previously reported rhythmic transcriptomes in cartilage and tendons, using identical tissue harvesting protocols and analysis algorithms, thus allowing cross tissue comparisons of the rhythmic transcriptomes[9,10]. This revealed a striking tissue specificity of rhythmic genes, with only 28 genes common to all three tissues (Fig. 2c; Supplementary Table 2).

These results show that breast tissue has an autonomous circadian clock, with a wide set of genes under circadian control.

**Mammary clocks are required for stem cell function.** To investigate the functional role of circadian clock genes, we chose the ClockΔ19 mutant mouse model, which has an inactive BMAL1/CLOCK complex and problems feeding pups owing to insufficient milk production[17,18]. This lactation phenotype becomes more apparent in the second litter (litter sizes of 2–3 pups in the mutant compared with ~10 in WT). Compromised stem cell function is responsible for this phenotype, which becomes more pronounced in the second, third or fourth litter[19]. As α6-integrin and Prkcε are rhythmic genes (Fig. 1d) and both are important for mammary stem cell function, we determined the role of the clock in breast biology by examining its effect on stem cell behaviour. The ClockΔ19 mouse has a severely suppressed mammary clock *in vivo* (Supplementary Fig. 1), which we hypothesized might compromise the capacity of progenitor cells to self-renew and generate functional mammary tissue.

Individual wild-type (WT) progenitor cells formed CD44-positive mammospheres in suspension culture, indicating that they have stem cell characteristics (Supplementary Fig. 2). Mammospheres arising from WT individual stem cells demonstrated rhythmic PER2::Luc oscillations, revealing the existence of autonomous clocks. In contrast, similar cells from ClockΔ19 mice had suppressed rhythmic oscillations (Fig. 3a,b). Although individual ClockΔ19 cells could form some primary mammospheres, their ability to do this was considerably reduced as revealed by Limiting Serial Dilution Assay (Fig. 3c). Furthermore in contrast with WT stem cells, almost no ClockΔ19 primary cells could form secondary mammospheres (Fig. 3d). These results show that circadian clock disruption compromised mammary stem cell, and that clocks are important for maintaining the biology of the mammary gland. Mammary gland phenotype has not been studied in other mouse models carrying mutations in different clock genes, which may be warranted in future studies.

**Aged mammary gland has a dampened clock.** Stem cell function deteriorates during the ageing of tissues[20–22] including the mammary gland. We therefore determined whether the mammary clock became dysregulated during ageing, as has been shown in other tissues[5–7]. We measured the amplitude of mammary circadian clocks in young and old mice. Long-term photon counting of PER2::Luc tissue explants using photon-multiplier tubes, revealed strong ~24 h rhythms in 3-month mice (Fig. 4a). In contrast, the robustness of PER2::Luc rhythms was markedly reduced in 24-month-old tissue. This occurred gradually, because the clocks were also suppressed twofold in glands from 12-month-old mice in comparison with those from 3-month-old animals (Supplementary Fig. 3). These results show that the mammary clock is suppressed during animal ageing.

Despite dampened clocks within whole tissues, primary mammary epithelial cells (MECs) isolated from 12-month-old mice retained robust clocks when individual cells were placed in tissue culture (Supplementary Fig. 4). Moreover, they displayed similar amplitudes to the MECs isolated from 3-month-old mice. These results show that there are comparable circadian oscillations within individual cells isolated from young and old mice. Thus, the effect of ageing on the mammary clock is not due to cell-intrinsic differences. Other tissues such as skin have a stiffer mechano-environment in old animals than young ones[23]. This suggests that the age-dependent dampening of mammary epithelial clocks might be caused by changes in the tissue mechano-environment.

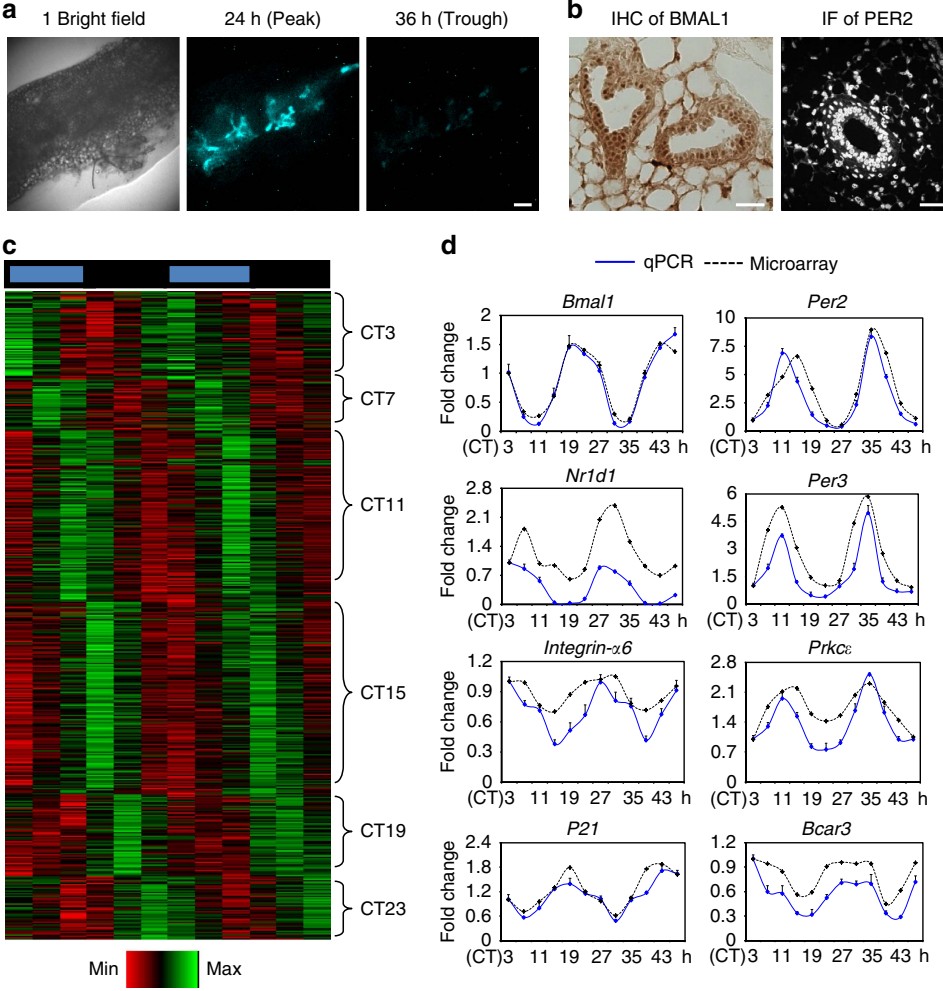

**Figure 1 | The circadian clock in mammary gland controls a tissue-specific set of rhythmic genes.** (**a**) Images of PER2::Luc bioluminescence (blue) in mammary tissue explants at peak and trough expression levels. $N = 3$ animals. Scale bar $= 200\,\mu m$. (**b**) IHC and IF staining of clock transcription factors BMAL1 (brown) and PER2 (white). $N = 3$ animals. Scale bar $= 50\,\mu m$. (**c**) Heat map of rhythmic mammary genes based on the circadian time-series microarrays. Genes are clustered according to timing of peak expression in circadian time (CT). Red, low expression; Green, high expression. (**d**) qRT-PCR validation of time-dependent expression of clock genes or clock-controlled genes, which are associated with progenitor/epithelial cell function (*Integrin-α6*, *Prkcε, P21* and *Bcar3*) in mouse mammary gland. $N = 3$ animals.

To assess whether the mammary extracellular matrix (ECM) became more rigid with ageing, we measured the stiffness of the periductal stroma at the nano-length scale using Atomic Force Microscopy. Aged mammary tissue had a significantly stiffer periductal stroma than that of young mice (Fig. 4b, Supplementary Fig. 5). Moreover, Picrosirius Red staining showed that fibrillar collagen within aged stroma was more highly organised than that in young tissue (Fig. 4c,d), similar to changes in aged skin (Supplementary Fig. 6). These results show that the ECM adjacent to mammary ducts *in vivo* becomes stiffer during ageing, correlating with suppressed mammary circadian rhythms that occur at this time.

**Extracellular microenvironment regulates the circadian clock.** Whether tissue stiffness controls circadian clocks has not previously been examined. To address this, we cultured purified MECs, which were isolated from PER2::Luc mice, under different mechano-chemical environments. This strategy avoids any complications that might arise from altered clock expression in other mammary cell types. Cells plated at high density in 3D culture form lumen-like acini[24], whereas those on 2D substrata

form monolayers (Fig. 5a). To determine whether the extracellular mechano-environment regulates the epithelial clock, we used bioluminescence photon counting and video imaging (Supplementary Fig. 7). There was more than sevenfold stronger circadian PER2::Luc amplitude in 3D cells compared with same number of cells cultured in 2D (Fig. 5b, Supplementary Movie 1, Supplementary Movie 2, Supplementary Movie 3). Moreover, the rhythmic mRNA levels of endogenous E-box containing clock genes (*Per2* and *Nr1d1*), and clock target gene (for example, α6-integrin) were stronger in 3D acini than 2D monolayer (Fig. 5c). Rhythmic expression of α6-integrin protein was also confirmed by immunofluorescence (IF) (Fig. 5d,e). Individual cells seeded at low density in 3D culture (stiffness of 30 Pa) showed robust circadian rhythms in PER2::Luc activity at the single cell level (Fig. 6a). However, those on 2D plastic dishes (stiffness of $> 100\,MPa$) had lower clock amplitude. These results show that the extracellular environment contributes to the strength of circadian activity.

To confirm a role for mechano-environmental stiffness on clock amplitude, MECs were cultured in a 3D alginate model in which we were able to control stiffness only[25]. Cells suspended inside similar hydrogels, which had either soft or stiff conditions

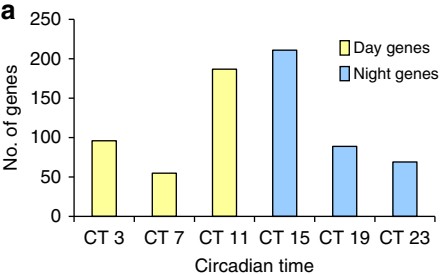

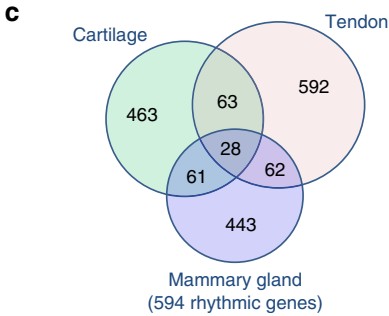

**Figure 2 | Analysis of the rhythmic mouse mammary gland transcriptome.** (**a**) Rhythmic mammary genes exhibited large-phase clusters at CT11 and CT15 (around the day/night transition). (**b**) GO-term analysis of circadian genes using DAVID. (**c**) Venn diagram compares the number of circadian genes of mammary gland, cartilage and tendon. These tissues were isolated from the same mouse strain under identical experimental conditions. The analysis algorithm and criteria were also identical. Note there is only ~10–15% overlap among tissues.

(as measured by pMLC levels and atomic force microscopy (AFM), Supplementary Fig. 8), showed a significantly suppressed circadian rhythm in the stiffer gels (Fig. 6b). Moreover, MECs cultured on 2D-substrata coated with different ECM proteins, such as laminin, collagen or fibronectin showed no difference in their clock activities, further supporting our hypothesis that the stiffness, rather than the composition, of the extracellular microenvironment controls clock activity (Fig. 6c). In addition to mammary epithelia, the circadian clocks of lung epithelial cells were also under mechano-dependent control, revealing that this mechanism of circadian control is not confined to mammary gland tissue (Supplementary Fig. 9). These results show that the cellular mechano-environment has a critical role in regulating circadian clocks in epithelia from both the mammary gland and lung, with increased clock strength in a softer microenvironment.

**Adhesion and cytoskeletal signals control the clock.** Mechanical forces of the extracellular microenvironment would have to be transduced through adhesion signalling systems and the cytoskeleton in order to impact on nuclear clock mechanisms. We therefore tested the role of integrin signalling by treating cells with shRNAs to the integrin-signalling protein vinculin, a major player in transducing mechanical cues into cells[26,27]. Cells with vinculin knockdown showed suppressed circadian rhythms, indicating the involvement of the mechano-sensing machinery in the regulation of cellular pace making (Fig. 7). In addition, we analysed whether disrupting the cytoskeleton with latrunculin B or cytochalasin D influenced the MEC clock in cells on the soft 3D matrix. Both of these compounds caused severe dampening of circadian oscillations (Supplementary Fig. 10A). These results show that cell–matrix interactions couple with the circadian clock directly via integrin signalling components such as vinculin, and that they require an intact cytoskeleton.

To further explore the role of the cytoskeleton, we determined whether intracellular tension driven by Rho/ROCK-mediated activation of actomyosin contractility controlled core clock transcription factors. Immunoblotting revealed that there were much higher levels of MLC phosphorylation in cells cultured on stiffer 2D ECM (Supplementary Fig. 8). Under these conditions, relaxing intracellular tension with the ROCK inhibitor Y-27632 reduced the levels of pMLC and the Young's modulus (Fig. 8a, Supplementary Fig. 8). Moreover, this treatment improved the circadian rhythm strength of PER2::Luc in MECs cultured within a stiff environment in a dose-dependent manner (Fig. 8b). Similar data were also obtained with another ROCK inhibitor, SR3677 (Supplementary Fig. 10). Mechanistically, ROCK inhibition significantly increased the ability of the CLOCK/BMAL1 complex to transactivate the E-box containing *Avp*::luc clock target gene reporter, but not the E-box mutant form *Avp*::luc reporter (Fig. 8c). These results show that inhibiting ROCK has marked effects on the circadian clock, arguing a role for the Rho signalling pathway.

To ratify further the involvement of Rho signalling, we expressed constitutive-active (Q63L-RhoA) and dominant-negative RhoA (T19N-RhoA) mutant constructs within MECs. Altered RhoA vectors caused a decrease (Q63L-RhoA) or an increase (T19N-RhoA) in circadian oscillations of PER2::luc in MECs (Fig. 9a,b). Also, knocking down endogenous RhoA in MECs significantly elevated the PER2::luc circadian amplitude (Fig. 9c,d). Finally, the activated form of RhoA (Q63L-RhoA) reduced transactivation of an E-box-containing reporter by CLOCK and BMAL1 when MECs were cultured in 3D, whereas dominant-negative RhoA (T19N-RhoA) enhanced expression of this E-box reporter in cells cultured on 2D ECM (Fig. 9e,f). These genetic manipulation studies show that the Rho signalling pathway has a critical role in regulating circadian clock amplitude in mammary epithelia, with reduced Rho levels or activity increasing the clock. The results support the suggestion that circadian clocks are mechano-sensitive.

**The mechano-sensing pathway influences mammary clocks *in vivo*.** Our experiments reveal that mammary clocks are sensitive to the stiffness of the cellular environment, and that they become suppressed during ageing. In old mice, intracellular proteins such as MLC, which sense tension provided by the extracellular environment, were phosphorylated at higher levels (Fig. 10a). To establish the contribution of the mechano-environment to the dampened circadian clock during ageing *in vivo*, we determined whether releasing intracellular tension could rescue the dampened clock of aged tissues. Fresh 3-month- and

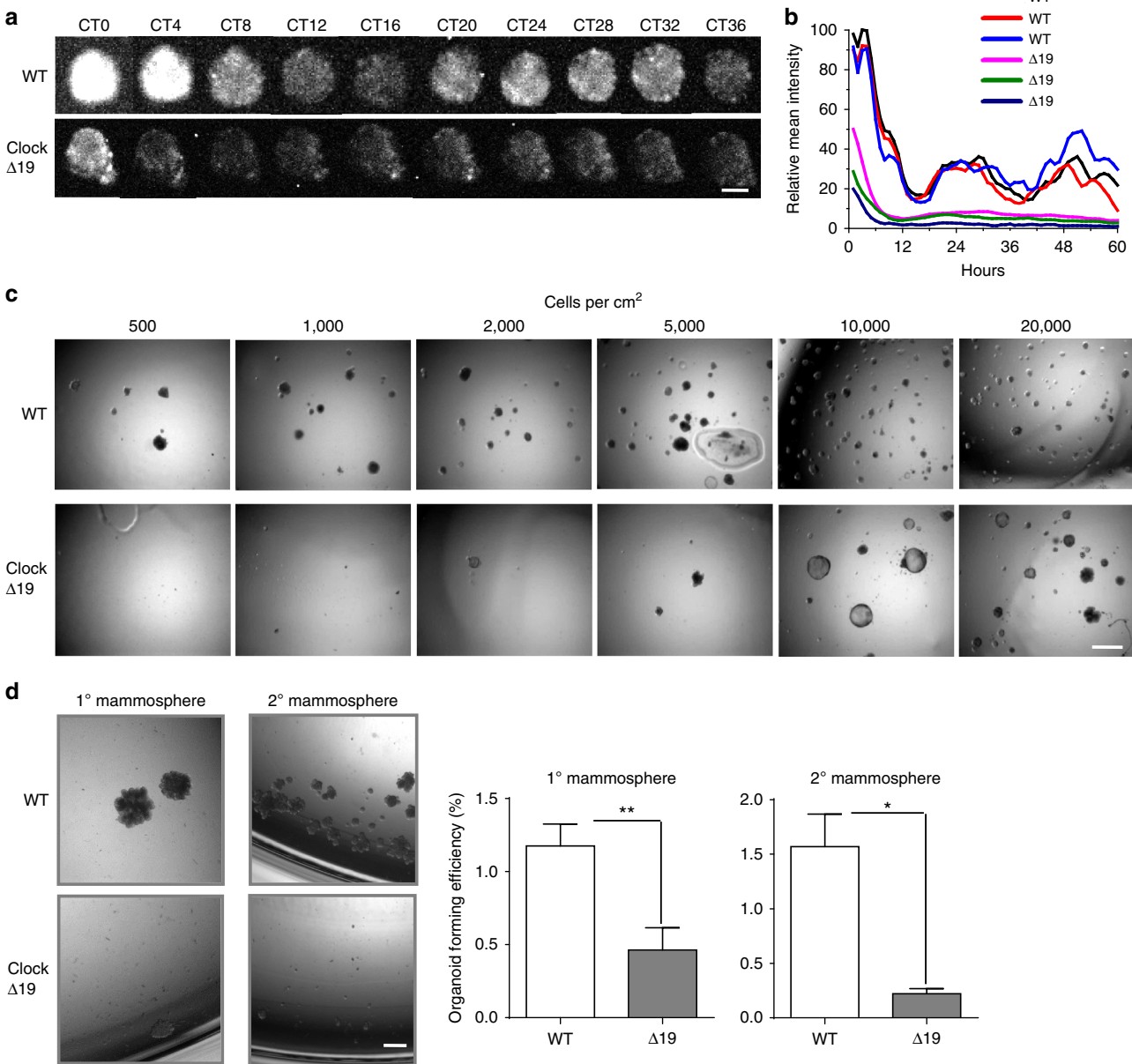

**Figure 3 | Mammary stem cells contain functional clocks that are required for their self-renewal capacity.** (**a**) Images of time lapse PER2::Luc bioluminescence in mammosphere structures isolated from WT or ClockΔ19 mutant female mice. Images represent sequential circadian time points after synchronisation with Dex. CT 0 (circadian time 0) denotes 24 h post Dex treatment. Note that owing to the weak signals from the ClockΔ19 mutant cells, the bioluminescence signals of the bottom panel were enhanced by 4 × to ease visualisation. N = 3 animals. Scale bar = 20 μm. (**b**) Quantification of mean intensity of PER2::Luc activity from individual wild-type or ClockΔ19 mammosphere structures. Three representative traces were plotted for each group. (**c**) Limiting Serial Dilution Analysis was performed using primary MECs isolated from wild-type or ClockΔ19 mutant mice. Numbers of single cells plated per cm$^2$ is shown at the top. N = 4 animals, scale bar = 100 μm. (**d**) Primary mammosphere assay was performed at a cell density of 2,000 individual cells cm$^{-2}$ from wild-type or ClockΔ19 mutant mice. Then a secondary mammosphere assay was performed by seeding 2,000 individual cells cm$^{-2}$ from dissociated primary mammospheres. Student's *t*-test, data were shown as mean ± s.e.m. *P < 0.05; **P < 0.01, n = 4 animals. Scale bar = 100 μm.

24-month-old mammary glands were treated with Y-27632, and their circadian rhythms were measured. We found that the ROCK inhibitor increased clock amplitude in older mammary tissue (Fig. 10b). This increase occurred to a small extent in tissues isolated from young mice, though the aged mammary gland showed an approximately twofold stronger induction. These results show that increased tissue stiffness contributes to the dampening of the circadian clock *in vivo*, providing a novel mechanism to explain how mammary clocks become altered during ageing.

## Discussion

Previous studies have reported that robust 24 h circadian rhythms are synchronized with daily environmental or metabolic changes[1–5], such as the light, feeding, temperature and hormones[4,5]. Our study now reveals that circadian clocks are also regulated by the stiffness of the extracellular micro-environment. Mechanistically we found that integrin signalling, actomyosin contractility and the Rho/ROCK pathway regulate the overall activity of the core circadian clock complex, CLOCK and BMAL1. Such tension-dependent transducers may, for example,

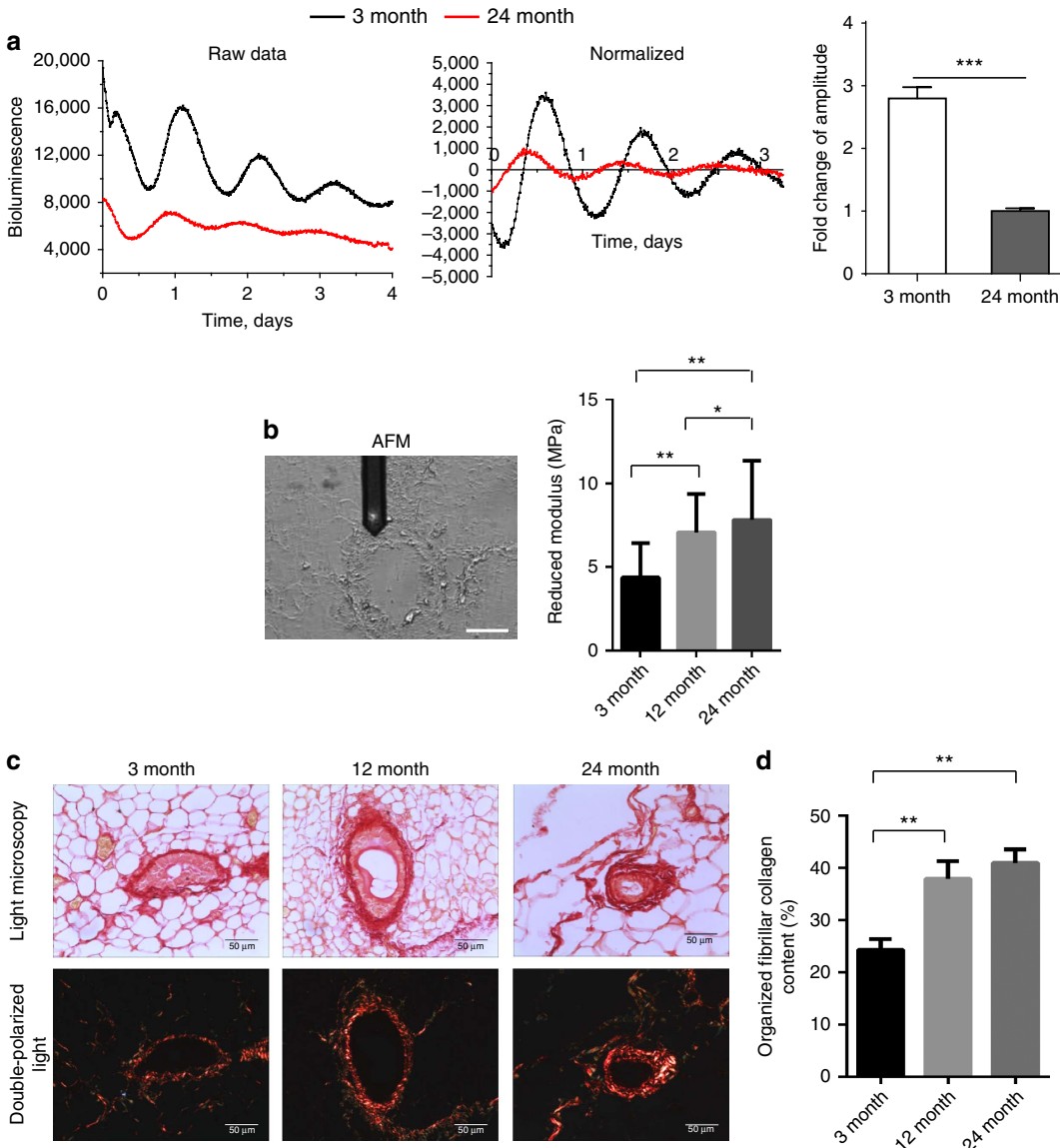

**Figure 4 | Aged mammary gland has a dampened clock and a stiffer mechano-microenvironment.** (**a**) Representative PER2::Luc traces from mammary tissues of either 3-month- (black) or 24-month-old (red) female mice. Right: fold change of amplitude was quantified (***, $P < 0.001$, $n = 5$ animals). (**b**) AFM measurements of periductal stiffness. Left: brightfield image of an unstained section under the AFM microscope with the periductal stroma clearly visible under the tip of the cantilever. Right, AFM quantification between 3-, 12- and 24-month-old mammary tissues. Data are shown as mean ± s.e.m., **$P < 0.01$, *$P < 0.05$, $n = 3$ animals, 1,200 replicates. Scale bar = 50 μm. The reduced modulus is related to the Young's modulus but includes corrections for the compliance of the indenter when indenting biological substrates with soft tips. (**c**) Picrosirius red staining in brightfield condition (top panel) and under double-polarized light (bottom panel) of 3-, 12- and 24-month-old mammary gland sections. $N = 6$ animals, scale bar = 50 μm. (**d**) Percentage of organized fibrillar collagen in (**c**) was quantified. One-way ANOVA, mean ± s.e.m., **$P < 0.01$.

control the activity of the clock transcription machinery via the YAP/TAZ or MRTF/SRF pathways, or possibly via cytoskeletal links to Nesprins and SUN proteins[28–32]. Indeed, in liver tissue and cultured fibroblasts, intracellular actin dynamics interact with the serum response factor to regulate circadian rhythms, whereas mechanical stimuli entrain *Drosophila* circadian locomotion behaviour via sensory receptors[33,34]. Given the links between ECM adhesion and intracellular signalling, our study reveals a novel pathway that controls the activity of molecular clock factors in epithelia.

Our transcriptome studies show that ∼600 rhythmic genes are under circadian control in the mammary gland, suggesting that circadian rhythms may have a fundamentally important role in this tissue. Of particular interest are the genes that have

previously been implicated in mammary stem cell function. Indeed, we showed that mice carrying mutations in the *Clock* gene have compromised the ability for stem cell renewal. Moreover, the size and number of mammospheres forming from MECs isolated from 24-month-old tissue was significantly lower than those from 3-month-old mice (Supplementary Fig. 11). Although ageing may simply reduce the numbers of stem cells[35], there might also be a link between age-related clock dampening and compromised stem cell function. Thus, disturbance of clock rhythms during ageing or in shift work may predispose breast tissue to diseases, and even cancer, as suggested by epidemiological evidence and GWAS studies[36–38].

In conclusion, we have established that extracellular stiffness and intracellular tension signalling provide a key pathway to

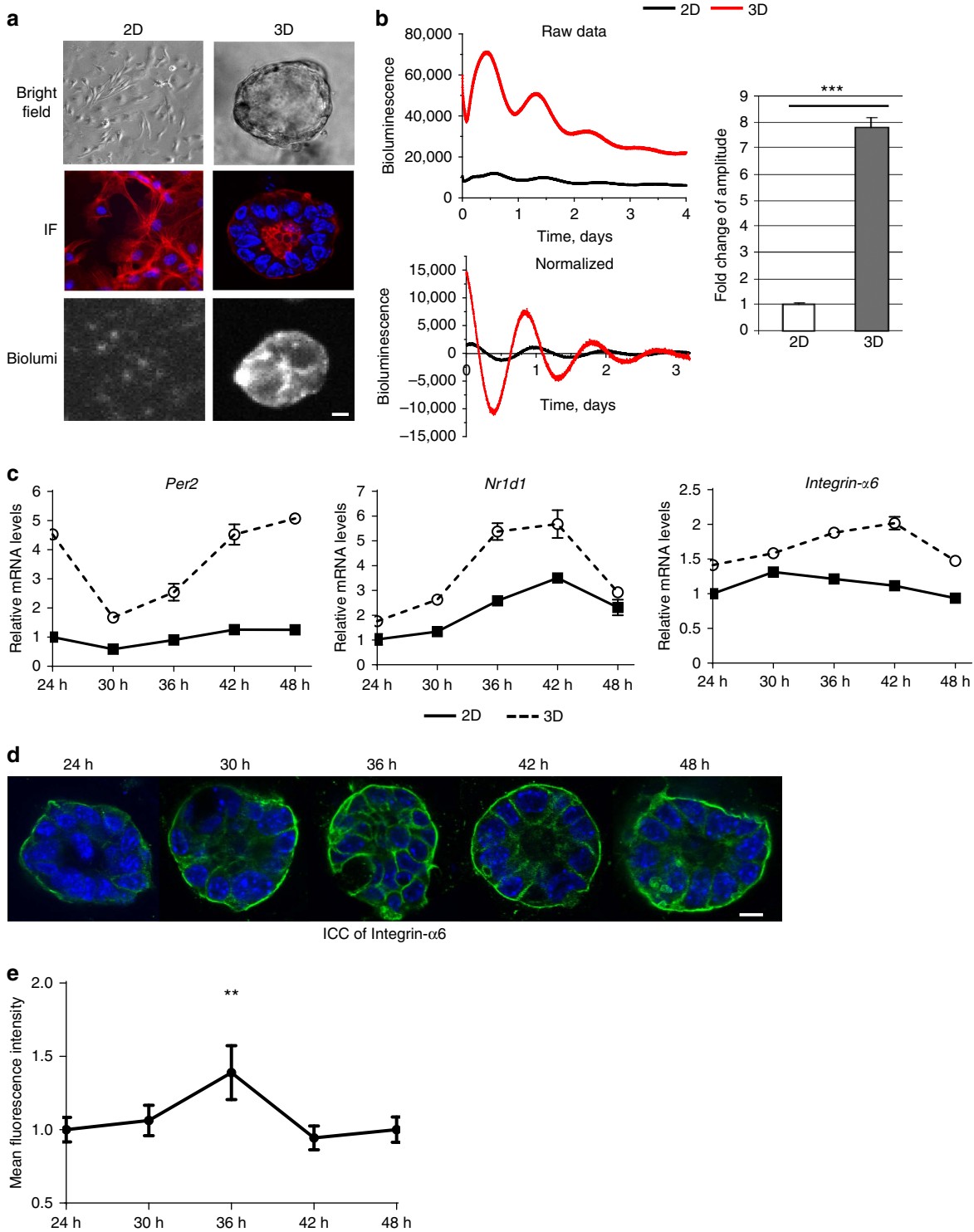

**Figure 5 | The extracellular microenvironment regulates mammary clock.** (**a**) Brightfield: primary MECs form monolayers in 2D collagen (2D) or acini structures in 3D Matrigel (3D), shown by phase contrast microscopy. $N = 3$ animals, scale bar = 20 μm. IF: red, β-Actin staining; blue, DAPI. Biolum: Similar cultures shown by bioluminescence staining of PER2::Luc. (**b**) Bioluminescence recordings and quantification of PER2::Luc activity from the same number of MECs cultured either on 2D or 3D. Student's $t$-test, mean ± s.e.m., ***$P < 0.001$, $n = 6$ animals. The fold amplitude in difference in activity is shown on the right. (**c**) The expression patterns of endogenous clock genes (*Per2* and *Nr1d1*), and clock target gene (for example, α6-integrin) in MECs cultured on either 2D or 3D, $n = 4$ animals. The mRNA levels were normalized to GAPDH and then to the 2D condition at 24 h. (**d**) Representative IF staining (blue, DAPI; green, integrin-α6) and (**e**) semi-quantification of integrin-α6 levels in MECs from WT mice cultured in 3D on Matrigel. Scale bar = 20 μm. **$P < 0.01$ for time-dependent expression, one-way ANOVA, $n = 3$ animals.

regulate the activity of circadian clocks in mammary epithelia. The downstream rhythmic genes identified in our study suggest that the clock may be involved in breast biology, for example, in

tissue regeneration following involution or in tumorigenesis. Indeed, the compromised self-renewal capacity of mammary progenitor cells in *ClockΔ19* mutant mice may contribute to

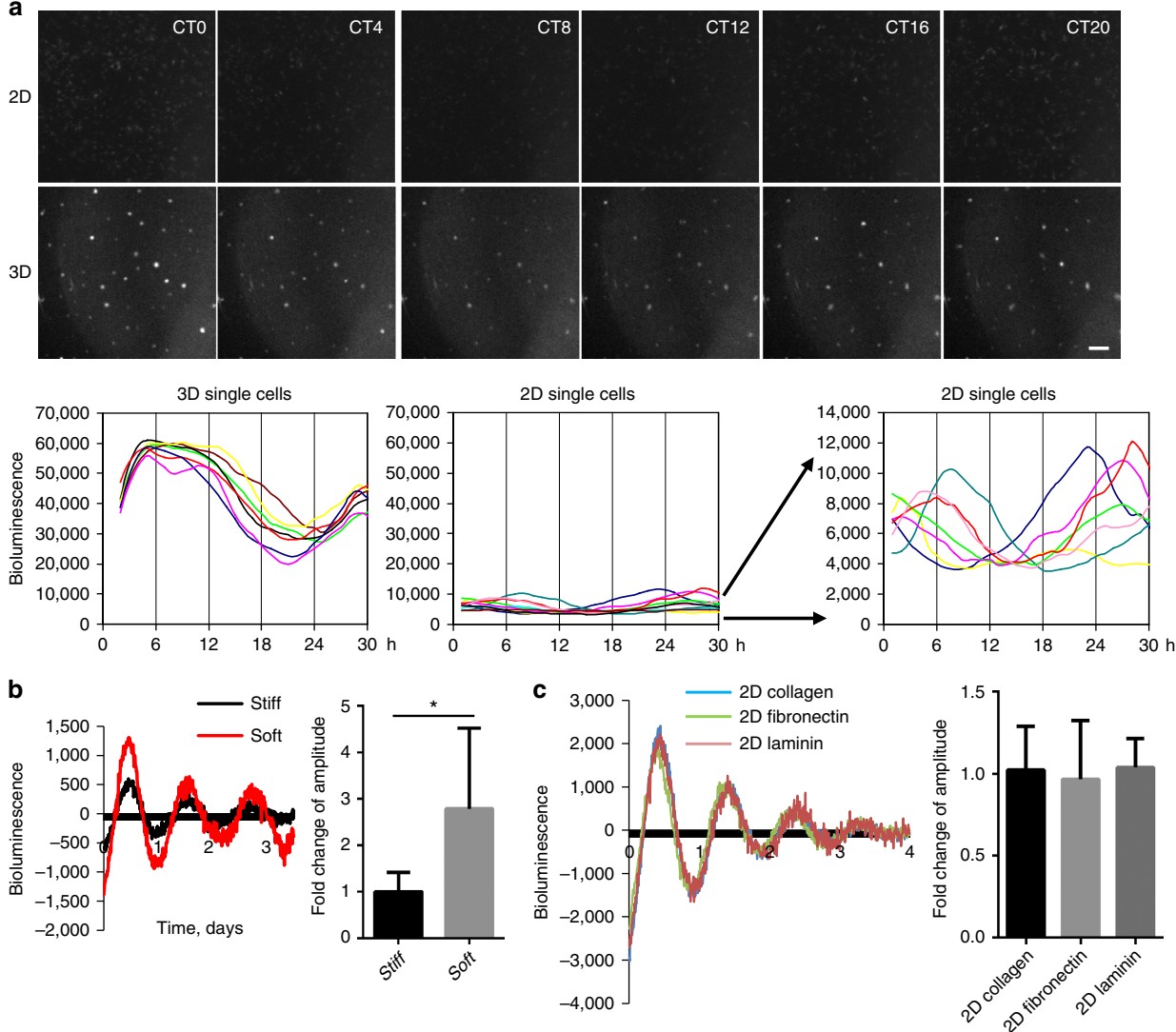

**Figure 6 | The mammary clock is cell autonomous and regulated by stiffness. (a)** Top: real-time bioluminescence imaging of primary MEC cells cultured as single cells at lower density, either on 2D or in 3D ECM. Bottom: quantification of bioluminescence signals from individual MECs cultured on either 2D or 3D. $N = 3$ animals, scale bar = 100 μm. **(b)** Primary MECs have stronger oscillations in a soft alginate matrix compared with a stiff alginate matrix. Left: PER2::Luc oscillations. Right: quantification of circadian amplitude. Student's $t$-test, mean ± s.e.m., *$P < 0.05$, $n = 4$ animals. **(c)** Representative PER2::Luc traces from primary MECs cultured on petri-dishes coated with collagen I (10 μg ml$^{-1}$), fibronectin (20 μg ml$^{-1}$) or laminin (10 μg ml$^{-1}$). Quantification shows no significant difference among these three conditions, one-way ANOVA, mean ± s.e.m., $n = 3$ animals.

nursing defects. Stiffening of the mechano-environment may also have a part in age-related dampening of clock rhythm, which could impact on a spectrum of downstream target genes that are involved in tissue homoeostasis and function.

## Methods

**Reagents and antibodies.** Dexamethasone (Dex) and Y-27632 were purchased from Sigma. The following antibodies were used in this study, BMAL1 (mouse monoclonal[9]), PER2 (rabbit polyclonal[39]), Phospho-Myosin Light Chain 2 (Ser19) (Cell Signaling #3675), Tubulin (Sigma), β-actin, CD44 Antibody (8E2F3) and Integrin alpha 6/CD49f antibody (from R&D systems). Alexa Flour 488 goat anti-rat was from Invitrogen; secondary antibodies conjugated to Cy5 were from Jackson Immunoresearch. mCLOCK and hBMAL1 expression plasmids were kind gifts from Dr Kazuhiro Yagita (Kyoto Prefectural University of Medicine, Japan). Antibodies used are cited in Supplementary Table. 3. Blots showing un-cropped scans used for the western blot figures are shown in Supplementary Fig. 12.

**Animal maintenance and tissue collection.** All experiments were conducted under the aegis of the 1986 Home Office Animal Procedures Act (UK). Mice were maintained on a standard maintenance chow under a 12-h light/12-h dark (12:12

LD) regimen. PER2::Luc and *Clock*Δ19 mice on a C57BL/6 J background were generated by Professor Joseph Takahashi. *Clock*Δ19 mice were subsequently bred with PER::LUC mice. For circadian tissue collections, 2-month-old female C57BL/6 J mice (Harlan Laboratories) were placed under 12:12 LD cycles for 2 weeks. before their release into DD. Animals were killed by cervical dislocation in complete darkness using an infrared viewer, and mammary gland tissues were harvested at 4-h intervals, beginning at 39 h after the start of DD. All tissues were either freshly used or snap-frozen in liquid nitrogen and kept at −80 °C until use. Circadian time (CT) corresponds to administration of light in the animal room. CT 0 indicates light-on, whereas CT12 indicates light-off.

**Mammary gland tissue explant cultures and bioluminescence recording.** Mammary gland tissues were dissected from either 10–12 weeks, or 22–24-month-old mice. The tissue explant was cultured on 0.4 μm cell culture inserts (Millipore), and bioluminescence was recorded in real time using photomultiplier tube (PMT) devices or a LumiCycle apparatus (Actimetrics)[39]. Baseline subtraction was carried out using a 24-h moving average. Cultures were also visualized using a self-contained Olympus Luminoview LV200 microscope and recorded using a cooled Hamamatsu Image EM C9100-13 EM-CCD camera. Images were obtained either once every hour for cells, or every 30 min for tissues, and results were combined in ImageJ. For Y-27632 treatment, mammary gland tissues were cultured under PMT

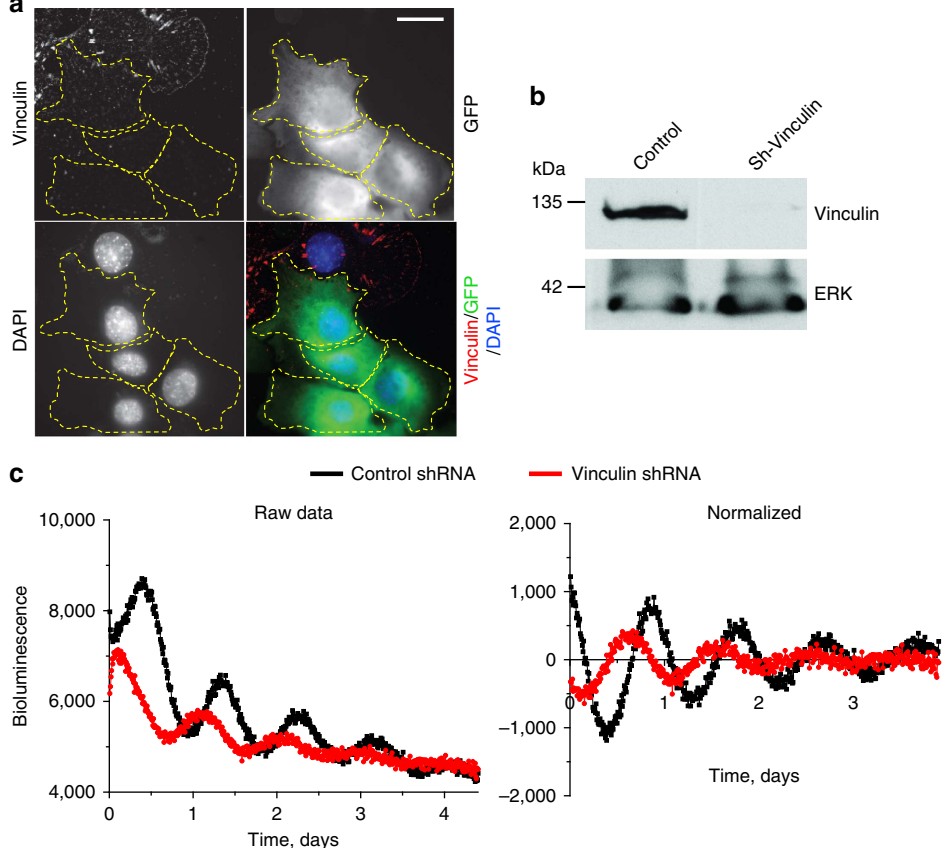

**Figure 7 | Adhesion signaling via vinculin is required for an effective cellular clock.** (**a**) Primary MECs expressing the GFP-sh-vinculin (outlined in yellow) have almost undetectable levels of vinculin. In contrast, cells not expressing GFP (green) have strong vinculin expression (shown by IF, red). Blue, DAPI. $N = 3$ animals, scale bar $= 20\,\mu m$. (**b**) Western blotting to confirm knockdown of vinculin in GFP-sorted MEC cells. Note: as a control, the levels of ERK were not affected by sh-vinculin. (**c**) Representative PER2::Luc traces from GFP-sorted MEC cells expressing lentiviral shRNAs against vinculin (red) or empty vector control (black). $N = 4$ animals.

recorders. After 2–3 days Y-27632 was applied, and the treatment agent was left continuously with the samples thereafter while the luminescence patterns were recorded for 2–4 days.

**Circadian time-series microarrays.** Frozen tissue was disrupted using FastPrep-24 lysing matrix tubes (MP Biomedicals). Extraction of RNA was carried out using RNeasy Mini Kit (Qiagen). Affymetrix Mouse430_2 GeneChips were run according to the manufacturer's instructions, and further analysis was performed as described previously[9]. In brief, CircWave Batch version 5 (provided by Dr Roelof Hut, University of Groningen) was used to fit a sine-wave with 24-h periodicity to each gene expression data set. Using known clock genes as a guide, a cutoff point of Rsq-value > 0.63 was arbitrarily assigned to determine circadian gene expression. JTKCycle (provided by Dr Michal Hughes, University of Missouri, St Louis) was also used to identify circadian transcripts. A Bonferroni-adjusted $P$ value of 0.01 was arbitrarily set as the cutoff for significance. To be stringent, genes identified using both methods were counted as positive. Validation of time-series arrays was carried out using TaqMan-based real-time PCR.

**IF and IHC.** Expression and distribution of proteins were visualized by indirect IF or immunohistochemistry (IHC). After 48 h of plating, cells were fixed for 10 min in PBS/4% (wt/vol) paraformaldehyde and permeabilized for 7 min using PBS/0.2% (vol/vol) Triton X-100. Non-specific sites were blocked with PBS/10% goat serum (for 1 h at room temperature) before incubation with antibodies diluted in PBS/2% goat serum (for 1 h at room temperature each). Cells were washed in PBS before mounting in either DAKO (DakoCytomation) for monolayers or prolong antifade (Invitrogen) for 3D acini[24]. Acini were visualized by confocal imaging. Images were collected on a Leica TCS SP5 AOBS inverted confocal microscope using a x63 Plan Fluotar objective. IF was performed on paraffin-embedded tissue and the luminal surface was detected with Cy5 (Jackson Immunoresearch) and imaged using confocal microscopy. IHC was performed on cryosections (10 μm), which were fixed with 4% formaldehyde. The standard avidin-biotin method was used, with

diaminobenzidine as the chromogen (Vector Laboratories). To confirm the antibody specificity, IHC was performed by skipping the specific primary antibody.

**Primary mammary cell culture and mammosphere assay.** Primary MECs were collected from 2–3-month-old virgin mice and cultured as described[40]. In brief, inguinal mammary glands were dissected and then digested via mechanical dissociation and enzymatic digestion with Collagenase A (30 mg ml$^{-1}$). Primary MECs were then sequentially centrifuged to enrich for and purify a mammary epithelial population. Cells were plated onto collagen I coated plastic petri-dishes for 2D monolayer cultures, basement membrane-matrix (Matrigel; BD Biosciences) to form 3D acini. Cells were cultured in growth media (Ham's F12 medium (Sigma) containing 5 μg ml$^{-1}$ insulin, 1 μg ml$^{-1}$ hy-drocortisone (Sigma), 3 ng ml$^{-1}$ epidermal growth factor, 10% fetal calf serum (Biowittaker), 50 U ml$^{-1}$ penicillin/streptomycin, 0.25 μg ml$^{-1}$ fungizone and 50 μg ml$^{-1}$ gentamycin). Mammosphere assay was performed as described[41]. In brief, dissociated primary MECs were seeded at a density of 500, 1,000, 2,000, 5,000, 10,000 or 20,000 cells cm$^{-2}$ in 12-well plates covered with 50 mg ml$^{-1}$ Poly-HEMA (Sigma) to prevent cell attachment. The cells were cultured with EpiCult-B Mouse Medium Kit (Stemcell technology), supplemented with 5% fetal calf serum (Biowittaker), 4 μg ml$^{-1}$ heparin (Stemcell technology), 10 ng ml$^{-1}$ epidermal growth factor, 20 ng ml$^{-1}$ bFGF (Sigma -F0291), 50 U ml$^{-1}$ penicillin/streptomycin, 0.25 μg ml$^{-1}$ fungizone, 50 μg ml$^{-1}$ gentamycin. To prevent cell aggregation, 5% basement membrane-matrix (Matrigel; BD Biosciences) was also added to be semi-solid. After 10 days in culture, mammospheres were collected and dissociated using 1:1 trypsin/DMEM solution following by passing through a 70 μm cell strainer. Single cells were then re-plated for second mammosphere assay at a density of 2,000 cells cm$^{-2}$.

**Alginate gels.** The 3D alginate gel system used in these experiments is described as before[25]. In brief, Alginate (Novamatrix, Norway) is mixed with BD Matrigel Reduced Basement Membrane (Corning, NY, USA) and freshly isolated, highly concentrated primary MECs from the PER2::Luc transgenic mouse. This mixture is then placed into a syringe, which is coupled to another syringe using a Luer lock

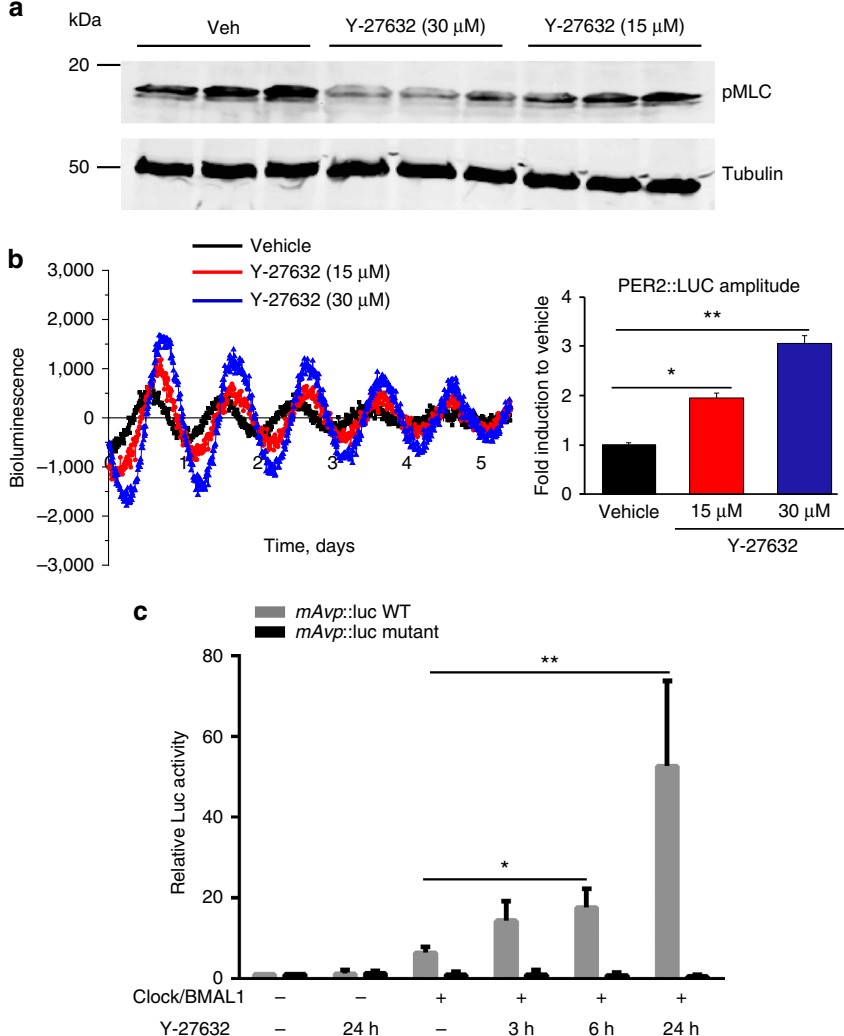

**Figure 8 | Relaxing the intracellular tension enhances clocks in mammary cells. (a)** Western blotting of pMLC expression in EpH4 mammary epithelial cells cultured in 2D with or without treatment of 15 or 30 μM Y-27632. $N = 3$ animals. **(b)** Left: representative bioluminescence traces of primary MECs cultured on 2D collagen, treated with either vehicle or different concentrations of Y-27632 (red, 15 μM; blue, 30 μM) at the beginning of the experiment. Right: quantification of fold change of PER2::Luc amplitude (based on first peak) induced by Y-27632 treatment. *$P < 0.05$; **$P < 0.01$, $n = 4$ animals. **(c)** Representative Luc assay result in EpH4 mammary epithelial cells. Clock and BMAL1 were co-expressed with Avp::luc promoter (wild-type or E-box mutant form) in the EpH4 cell line. Cells were then incubated with Y-27632 (30 μM) for different durations as indicated. Following the treatment, cells were lysed for luciferase assay. Data were normalized to β-galactosidase reporter gene and expressed relative to pCDNA3 control. Student's $t$-test, mean ± s.e.m., *$P < 0.05$; **$P < 0.01$ versus Clock + BMAL1 group, $n = 3$ replicates.

coupler (ValuePlastics). The second syringe is filled with a mixture of blank DMEM and calcium sulphate solution. The concentration of calcium sulphate solution determines the stiffness of the gel, as it catalyses cross-linking between the polymers. The two solutions are then mixed together by rapidly depressing the syringes, and then expelled into 35 mm dishes that were pre-coated with Matrigel. Gels were then left to set in an incubator. After 30 min, normal growth media was added and gels were returned to the incubator. Two days later, cells were treated with Dexamethasone (a known clock synchronising agent acting on the GRE elements in the promoters of *Period* genes) for 1 h, then changed to recording medium and placed in the Lumicycle.

**Lentivirus delivery of shRNAs in primary MECs.** The lentiviral shRNA vector, pVenus, was provided by Didier Trono (University of Geneva, Geneva, Switzerland).shRNA for mouse vinculin was designed with shRNA design tool (Open Biosystems). The target sequence for mouse vinculin is 5′-CGAGATCATTC GTGTGTTA-3′. A BLAST search did not reveal any other target sequences in mouse. Doubled-stranded oligonucleotides were cloned into the lentiviral transfer vectors pVenus. For lentivirus production, the transfer vectors were co-transfected with the envelope plasmid pMD2G and the packaging plasmid psPAX2 into HEK 293 T cells using PEI reagent. Media were replaced after 8–10 h. In total, 10 ml viral supernatant was harvested 48–60 h after transfection, passed through a 0.45-μm

filter, and further concentrated by centrifugation at 25,000 r.p.m. at 4 °C for 2.5 h. Viral pellets were re-suspended in 0.1 ml fresh DF12 medium. For lentiviral transduction, primary MEC cells from PER2::Luc mice were grown to 80% confluency in 35 mm dishes. Lentiviral infection was performed by adding lentiviral particles directly to cells and incubating for 8 h. The infected cells were cultured for 48 h to ensure turnover of pre-existing vinculin. Pure population of infected cells was enriched by FACS sorting the Venus positive cells, followed by real-time recording of bioluminescent activity using Photomultiplier. Knock-down of vinculin was confirmed by IF (before sorting) and western blotting (after GFP sorting) using an anti-Vinculin antibody (clone VIN-11-5, Sigma-Aldrich).

**Primary lung epithelial cell isolation.** The cells were isolated as described in previously published method[42]. In brief lungs from PER2::Luc mice were perfused with PBS and digested using Elastase solution (Sigma-Aldrich, 4 U ml$^{-1}$ in HBSS). Then lung lobes were minced and incubated with DNase I solution for 15 min at 37 °C. The cells were passed through a 70 μm cell strainer (BD#352350), and then resuspended in cold RBC lysis buffer (eBioscience #00-4333-57) to remove the red blood cells. Live cells were counted based on trypan blue exclusion. These cells were then plated into 3D basement membrane-matrix (Matrigel; BD Biosciences) before real-time recording of bioluminescent activity.

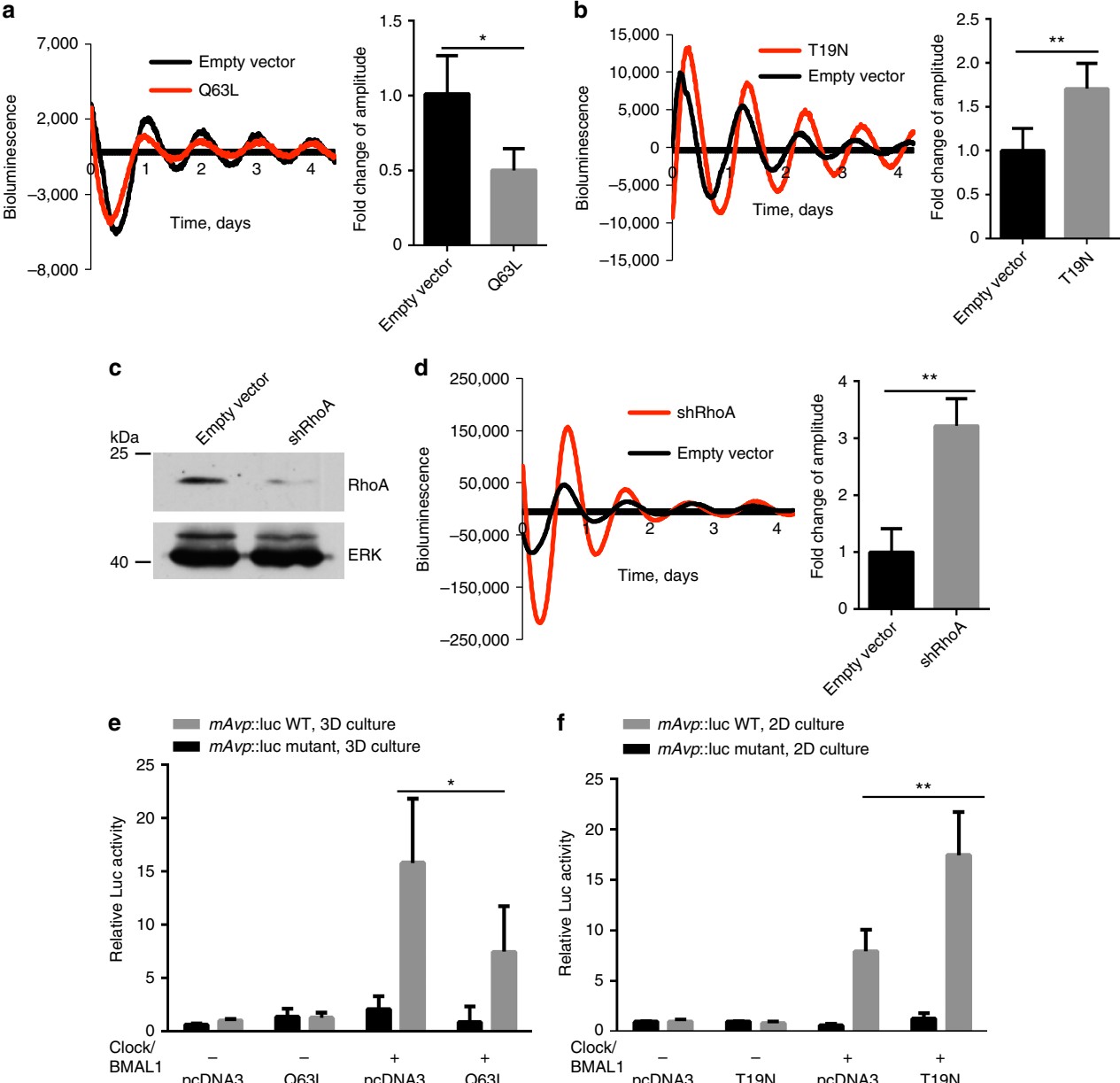

**Figure 9 | Genetic manipulation of Rho signalling regulates the MEC circadian clock. (a)** Representative PMT traces from EpH4 cells in 2D/3D following transfection with RhoA plasmid. Co-expression with a constitutively active RhoA (Q63L) in EpH4 cells cultured in 3D produces a twofold reduction in the amplitude of circadian oscillations. Black, transfection with empty vector (pcDNA3), control; Red, transfection with Q63L plasmid. Student's *t*-test, mean ± s.e.m., *$P < 0.05$, $n = 3$ replicates. **(b)** Co-expression with a dominant-negative form of RhoA (T19N plasmid, as shown in red) in EpH4 cells cultured in 2D increases the amplitude of circadian rhythm by approximately 1.7-fold. Black, transfection with empty vector control. Student's *t*-test, mean ± s.e.m., *$P < 0.05$, $n = 3$ replicates. **(c)** Knockdown of shRhoA in EpH4 cells confirmed by western blot. $N = 3$. **(d)** Representative PMT traces from EpH4 cells in 2D following lentiviral transduction of an shRhoA vector (red) or an empty vector, control (black). Knockdown of RhoA in EpH4 cells in 2D increases the robustness of the rhythm. The amplitude of the oscillations is increased threefold following knockdown of RhoA. Student's *t*-test, mean ± s.e.m., **$P < 0.01$, $n = 3$ replicates. **(e)** Representative Luc Assay in EpH4 mammary epithelial cells in 3D culture. Co-expression with a constitutively active RhoA vector in EpH4 cells cultured in 3D reduces transactivation of E-box targets. **(f)** Co-expression with a dominant-negative form of RhoA in EpH4 cells cultured in 2D increases activation of E-box targets. Data were normalized to β-galactosidase reporter gene and expressed relative to pCDNA3 control. Student's *t*-test, mean ± s.e.m., *$P < 0.05$; **$P < 0.01$ versus Clock + BMAL1 group, $n = 3$ replicates.

**Functional luc assay.** *mAvp*::luc plasmid and its E-box mutant form, and luc assay were described previously[39]. In brief, EpH4 cells were seeded at 50% confluency in 12-well plates. Cells were transfected with mixtures containing 0.5 μg reporter construct, 0.5 μg β-Galactosidase expression vector, 0.5 μg of BMAL1 and CLOCK expression vectors, 0.5 μg carrier pcDNA or a variation whereby some of the vectors are excluded, using PolyFect Transfection Reagent(Qiagen). The following day, cells were treated with 30 μM Y-27632 at 3, 6 and 24 h. Luciferase activity was recorded using the Promega Luciferase Assay System, on an Orion L Microplate Luminometer with Simplicity 4.2 Software. Data were analysed using a Student's *t*-test in GraphPad Prism 6.

**Real-time qRT-PCR.** RNA was extracted from either tissues or cells using the Qiagen RNeasy purification system. cDNA was prepared using a High Capacity

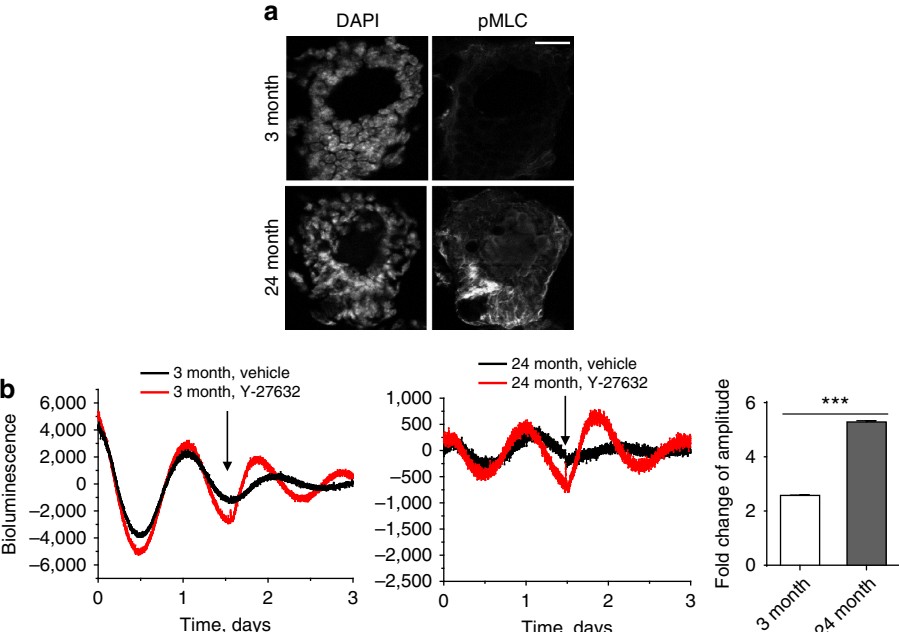

**Figure 10 | The mechano-sensing pathway influences the mammary clock *in vivo*.** (**a**) IF staining of pMLC expression in either 3-month or 24-month-old mammary gland tissue. $N = 3$ animals, scale bar $= 50\,\mu m$. (**b**) Representative PER2::Luc bioluminescent traces of mammary tissue explants from 3-month or 24-month-old mice, treated with Y-27632 ($100\,\mu M$, black arrow). Red, Y-27632 treatment; black, control treatment. Fold change of amplitude was quantified based on first peak following the treatment. Student's *t*-test, mean ± s.e.m., ***$P < 0.001$. $N = 3$ animals.

RNA-to-cDNA Kit and analysed for gene expression using quantitative real-time PCR with TaqMan (Applied Biosystems) chemistry. Primer/probe mixes were ordered from Applied Biosystems. For clock genes, the primers are as follows, *Per2*: Mm00478113_m1; *Bmal1/Arntl*: Mm00500226_m1; *Per3*: Mm00478120_m1; *Nr1d1/Rev-erbx*: Mm00520708_m1. For the mammary specific rhythmic genes, the primers are as follows, *Bcar3*: Mm00600213_m1; *P21*: Mm04205640_g1; *Itga6*: Mm00434375_m1; *Prkcε*: Mm00440894_m1. Results were normalized to the values for *Gapdh* expression, using the $2^{-\Delta\Delta Ct}$ method.

**Picrosirius red staining and analysis.** Wax sections were stained with Picrosirius Red. By incorporating the birefringent properties of fibrillar collagen, we were able to devise a semi-quantitative method of analysing the organisation of collagen molecules in the ECM. Using ImageJ 1.48, areas of interest were isolated using the freehand tool. The basement membrane that surrounds the mammary ducts and the dermoepidermal junction (DEJ) in the skin were chosen as the primary region for analysis[43]. In the brightfield image, a trace was drawn around the specified areas, and the area of this trace was recorded. The trace outline was then copied onto the corresponding polarised image. The selection was then inversed and cleared. This resulted in the deletion of the remainder of the image, leaving only the collagenous region desired. Finally, a red colour threshold was applied and the area of the image that passed the threshold was selected. The area captured by these selections was measured, and the value divided by the total area initially captured from the brightfield trace. This analysis was performed over serial paraffin sections. Statistical analyses of the percentage values obtained were performed using a one-way ANOVA in GraphPad Prism 6 analysis package.

**AFM.** AFM on frozen sections were performed using $5\,\mu m$ thick cryosections as previously described[44,45]. The reduced modulus was calculated which is related to the Young's modulus, but includes corrections for the compliance of the indenter and is more commonly used when indenting biological substrates with soft tips. In brief, the periductal stroma and the DEJ were identified by comparing the unstained tissue with serial sections stained for Picrosirius Red. A Bioscope Catalyst (Bruker, Coventry, UK) was used, mounted onto an Eclipse T1 inverted optical microscope (Nikon, Kingston, UK) that was fitted with a spherically tipped cantilever (nominal radius and spring constant of $1\,\mu m$ and $3\,Nm^{-1}$, respectively: Windsor Scientific Ltd., Slough, UK). The local reduced modulus was determined for each of 400 points in a $25 \times 25\,\mu m$ region, indented at a frequency of 1 Hz with lateral spacing of $1.25\,\mu m$. The extend curve was used in conjunction with a contact-point-based model to calculate the reduced modulus for each indentation. For each biological sample, three $25\,\mu m^2$ regions, and hence 1,200 force curves, were collected. *Post hoc* analyses of force curves were performed using Nanoscope Analysis v 1.40 (Bruker), whereby a baseline correction was applied to each curve before a force fit was applied using the Herzian (spherical) model and a maximum force fit of 70%. Once all 400 force curves had been generated, quality control was

applied, whereby any force values falling more than two s.d.s away from the mean value were discarded in order to account for failed indents. In general fewer than 10% of force curves were excluded. For each mammary section, three ducts were isolated, and within each duct, three areas of periductal stroma were scanned. In the skin sections, three areas along the length of the DEJ were examined per section. Three mice per group were evaluated for both skin and mammary tissues. The data were analysed using GraphPad Prism 6 analysis software.

**Statistical analysis.** Data were evaluated using Student's *t*-test, Mann–Whitney *U*-test, or one-way ANOVA with Tukey test as indicated. Results were from at least three independent experiments, and shown as mean ± s.e.m. Differences were considered significant at the values of *$P < 0.05$, **$P < 0.01$ and ***$P < 0.001$.

**Data availability.** The authors declare that the data supporting the findings of this study are available within the article and its Supplementary Information files and from the corresponding author upon request. Microarray data have been deposited in Array Express under Accession code E-MTAB-5330.

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

## Acknowledgements

We thank professor Joseph Takahashi for PER2::Luc and ClockΔ19 mice. We thank Dr Paul Potter (MRC Harwell) for providing us aged C57BL/6 female mice. We thank Dr Leo AH Zeef from Bioinformatics Core Facility for his help with the microarray data analysis. We thank Professors Cay Kielty, Rob Lucas and Andrew Loudon for insightful comments and critical reading of the manuscript. This work was supported by a Career Development Award (G0900414) and a Centenary Award from the Medical Research Council, UK to Q.-J.M; funding from the Wellcome Trust, UK to the Wellcome Trust Centre for Cell-Matrix Research (088785/Z/09/Z); funding from the BBSRC for JW (DTP PhD studentship); and funding for purchase of the Olympus Luminoview LV200 microscope (Olympus) and Hamamatsu ImageEM C9100-13 EM-CCD camera (BB/J003441).

## Author contributions

Q.-J.M. and C.H.S. designed the experiments. N.Y., J.W., V.P.-V., P.W., S.O., J.M., N.G., A.H., J.C. and Q.-J.M. conducted the experiments and acquired the data. N.Y., J.W., V.P.-V., P.W., S.O., J.M., N.G., A.H., J.C., C.H.S. and Q.-J.M. analysed the data. Q.-J.M. and C.H.S. wrote the manuscript.

## Additional information

**Competing financial interests:** The authors declare no competing financial interests.

