## [Peer Review File · Nature Communications]

Referee #1

In this manuscript, the role of the external physical environment as a major influence on circadian regulatory mechanisms in mammary cells has been examined. The study starts off with transcriptomic analysis that reveals a large number of genes regulated in a circadian manner. Several are validated by various means. The presence of example genes associated with mammary stem cell function led the authors to examine the relationship between circadian regulation of a reporter construct in wild-type and ClockDelta19 cells with the ability of isolated mammary cells to form mammospheres in 3-D, a response related to the presence of stem cells in a population. The relationship between the ClockDelta19 sensitivity in both these assays is taken to indicate a causal connection. Next, explant mammary tissue from young and aged mice was examined for the associations between circadian rhythms, tissue elasticity and collagen deposition/organization, which indicated a relationship between increased tissue stiffness and less circadian cycling. To examine the mechanistic basis for these correlations, mammary epithelial cells (MEC) were isolated and used in varying conditions, including plating in 2-D vs 3-D environments, or under varying stiffness matrices. These experiments indicated that sensing of external rigidity leads to cellular tension, mediated via ROCK phosphorylation of MLC, which negatively affects circadian patterns.

The manuscript has been strengthened by the additional data compared to the first submission. The focus of the manuscript on how the external environment affects circadian regulation via cellular tension is well supported. However, there are still a number of parts of the manuscript that are not effectively linked. The data in Figure 1 shows that there are clear patterns of gene expression changes regulated in a circadian manner is not actually relevant to the rest of the manuscript in meaningful way. These data haven't been linked to difference observed between young and aged mammary tissue, for example. Nor has this information been used to determine how circadian regulation of gene expression contributes to mammary stem cell biology. As a result, these results are essentially superfluous and could be removed with no significant impact on the manuscript. The link from circadian regulation of gene expression to a possible role in mammary stem cell regulation to mechanosensing is unclear and has not been effectively linked with the rest of the manuscript.

Similar to Figure 1, the data in Figure 2 could be removed with little to no impact on the rest of the paper.

Please note that Figure 1 is the first ever genome-wide characterization of the rhythmic clock target genes (output) that are regulated over the circadian 24-hour cycle in breast, which will be immensely informative for the field to investigate the function of the mammary gland circadian rhythms. They are crucial within this paper because they set the stage for the rest of the paper. We never intended to compare RNA array expression in young vs old mice, as this is neither relevant nor practical (we would need many old mice) to the rest of the paper.

The mechanosensing studies were designed to investigate the input pathway to the mammary epithelial clock. We are not necessarily expecting any rhythmic genes to be linked to the regulatory (input) pathway. However, as the very first detailed study of the mammary gland circadian function, we feel the transcriptome data adds significantly to the research field.

Please note that Figure 2 shows that circadian clocks are essential for stem cell function within the mammary gland which could go some way to explain the nursing defects of the Clock mutant phenotype. This is a completely new finding that has not been published by

environment impacts on circadian regulation has been strengthened and is quite convincing as far as they go. One question that the findings provoke is whether an ectopic Rho or ROCK signal would over-ride the circadian patterns observed in MEC cells plated in conditions that allow it (e.g. 3D). Another question is what mediates the effect of cellular tension on the circadian cycling.

This referee has raised a new question about expressing ectopic Rock. As requested, we have now done additional experiments with constitutively-active or dominant-negative forms of ectopic RhoA. We have also performed shRNA knockdown of endogenous RhoA. Together these experiments back up the Y-27632 inhibitor studies. We have included the data in additional supplementary figures, S14, S15, and S16. To explain the studies, we have also added additional text within the manuscript. This is on page 7, as follows:

“These results show that inhibiting ROCK has dramatic effects on the circadian clock, arguing a role for the Rho signalling pathway. To ratify further the involvement of Rho signalling, we expressed constitutive-active and dominant-negative RhoA mutant constructs within MECs. An activated form of RhoA reduced transactivation of an E-box-containing reporter by Clock and BMAL1 when MECs were cultured in 3D, while dominant-negative RhoA enhanced expression of this E-box reporter in cells cultured on 2D ECM (Fig. S14). Also, altered RhoA vectors caused a decrease (Q63L-RhoA) or an increase (T19N-RhoA) in circadian oscillations of *Per2::luc* in MECs (Fig. S15). Finally, knocking down endogenous RhoA in MECs significantly elevated the *Per2::luc* circadian amplitude (Fig. S16). These genetic manipulation studies confirm that the Rho signalling pathway has a critical role in regulating the amplitude of circadian clock in breast epithelia.”

Given that ROCK inhibitor has been well established as being very important for in vitro stem cell culturing, that circadian cycling is apparently necessary for mammary stem cell growth in vitro (Figure 2), and ROCK inhibitor promotes sustained circadian cycling (Figure 5C), is the ability of ROCK inhibitor to promote mammary stem cell culturing a reflection of this positive effect on the circadian regulation of gene transcription? This question seems to be a central one to linking the mammary stem phenomena with the latter part of the manuscript.

As requested, we have now done some studies where we cultured mammary stem cells in the presence or absence of a ROCK inhibitor. The results are that inhibiting ROCK is actually necessary for efficient mammary stem cell formation (data available upon request). This observation strongly indicates that ROCK inhibition is a prerequisite for stem cell survival - note though that the suggested experiment would not actually tell us anything on the effect of circadian cycling.

Minor issues

1. All photomicrographs should have scale bars (Figure 1A, 1B, etc.).

We've added scale bars to all the figures.

2. The data in Figure S6 would benefit from being plotted as normalized data as in Figure S5, with similar analysis of amplitudes of each data set.

As requested, we've added normalised data plots to FigS6.

3. At the bottom of MS page 3, it would be better to suggest that the effect of ageing on

the mammary clock might be due to changes in the tissue mechano-environment, rather than saying that it is likely. At this point in the manuscript, this statement is quite a leap of faith.

As requested, we changed 'likely' to 'might be'.

4. Define CT used for labelling Figures 1C, 1D, etc in the manuscript text, with information about the units of this measurement.

Please note that the description of CT (= circadian time, in hours) is already in the legend of Fig 1C.

5. X-axis in Fig S1B should be labelled.

Please note that the x-axis in Fig S1B (= Gene counts) is already the Figure.

6. MS page 5, Figure S10D shows the effect of Y27632 on Young's modulus, not pMLC as indicated in the manuscript.

It is both, actually: both pMLC and Young's modulus. We changed the text slightly.

Referee #2

I appreciate that the authors of NCB-S29095B made a credible effort to address this reviewer's concerns. However, even with the additional data, the conclusions are still not supported.

We thank the authors for cultivating our knowledge regarding the AFM measurement length scale by providing a figure from Akhtar et al., (2011) Materials Today. Indeed, several groups have been using the same technique to measure breast epithelial and stromal stiffness, and they all have found it to be at a maximum of 2.5kPa (Lopez et al. Integrative Biology 2011), which sharply contrasts with what is reported here. Not sure why the discrepancy, but unfortunately the onus is on the the authors of S29095B to address this.

As requested, we have revisited the AFM question. The initial concerns raised by the referee cited a paper that reported breast stiffness at 2.5 kPa. In that paper they used a silicon nitride pyramidal cantilever with a spring constant of 0.06N/m that was customised to have a 5- μ m borosilicate glass sphere attached as the tip. The authors measured 100 points on 20- μ m thick sections over a 90- μ m x 90- μ m square region that spanned a mammary duct. The area of analysis included epithelia, stroma, adipose and basement membrane. Importantly, such large regions of tissue have a dramatic effect on the final value stated for 'Breast Stiffness'.

By contrast in our study, we examined the immediate micro-environment surrounding ductal epithelia. We focused on the basement membrane, and collagen-rich area revealed by Picrosirius Red staining. For these reasons we therefore used a spherical tip of 1- μ m in diameter, and measured 400 equally spaced points over a 25- μ m x 25- μ m square area. This is quite different to the method mentioned above, and we have measured a much smaller area of tissue.

To reconcile these differences, we have now repeated part of our study using a large tip and a cantilever with a softer spring constant, and also an indentation paradigm similar to previous studies, to mimic the experimental conditions as close as possible to that of the cited publication. New sections of tissue were cut at 20- μ m thick (as in the cited paper), as opposed to the 5- μ m sections that we used initially. Interestingly, we found that the absolute

values for Young's Modulus were roughly 200-fold softer than with the smaller radius tip and softer cantilever, which is in line with the cited paper (see Fig 1 and Table 1 below).

Figure 1. Atomic Force Microscopy recordings of young mammary tissues under two distinct AFM parameters that differ in tip radius and spring

Tip size	Spring Constant	'Breast Stiffness'
1 μm	3 N/m	4.5 MPa (Our manuscript)
10 μm	0.2 N/m	28 kPa (new experiment)
5 μm	0.06 N/m	2.5 kPa Lopez et al., 2011)

Table 1. Absolute values recorded for 'Breast Stiffness' are significantly different in 3 independent experiments.

The values obtained using our original tips are not unusual for AFM measurements. In fact, our group in Manchester used the same set of tips to measure the stiffness of the area surrounding ducts in human breast tissue (McConnell et al., 2016 *Breast Cancer Research*). There, the observed values of Reduced Modulus ranged from 200 kPa to 1.5 MPa, which are more comparable to the values seen in our current study, which is in mouse mammary gland.

Changes in cantilever length, spring constant, tip radius, tip shape, tip coating and buffer conditions for fluid indentations can all affect the absolute values recorded using AFM. It is important to choose the right parameters for each experimental design and equally important to state these in the methods section of publications, as we have done in our manuscript.

Because absolute values are so dependent on the afore-mentioned variables, comparing readings from separate publications is not appropriate. We feel that the conclusions within a publication must therefore reflect the *relative* changes in stiffness between the tissue areas measured within the same system, rather than the actual values.

Other issues

In all the references cited, the circadian rhythm attenuation in amplitude is attributed to tissue explant. This is not due to cells losing synchrony, that is not how it would look mathematically. The authors need to find a better explanation for this decrease in their mouse model (Fig 3A).

This experiment showed that mammary tissue from old mice had a much lower level of clock activity than that from young mice. We did other experiments to argue that the differences were caused by changes in the tissue mechano-environment. Please note that we didn't discuss anything about altered synchrony, indicating that this criticism is not relevant.

The authors should more precisely validate the stiffness of the alginate gels used. In the rebuttal, they indicate that the alginate soft stiffness is around 30Pa and the more rigid stiffness is around 300Pa. It does not make any sense why the authors used this range of stiffness, since they measured a stromal stiffness in the order of Mega-Pascal.

Unfortunately, this comment about the alginate gel is a misunderstanding. The alginate gel is a new culture model that has hardly been used by anyone else. We have merely used this system to show that cells in different relative stiffness microenvironments have different clock levels. The reason to do this was to use a different culture model, as proposed by the first Reviewer's original comment, to strengthen our overall arguments hugely. Please note that we used the system to show that different stiffness impacts on cell phenotype, but we did not use it to mimic the precise stiffness of tissues.

Fig S14, the authors should repeat this experiment using CD44 stem cells, like they used in Fig. 2. They should show the proportion of the CD44 cells as a function of age to make the connection stated in the discussion.

This comment raises new and additional experiments on assessment of stem cell activity (using CD44) in aged mammary gland. However, the Jackson et al NCB paper that we cited (Ref35) has already nicely demonstrated this point. They have shown that "We followed the cellular composition of the mammary gland through the four stages of the adult mouse lifespan: post-puberty, mature, middle-aged and old. Flow cytometry indicated that basal, luminal progenitor and luminal differentiated cell compartments were consistently maintained in the WT." "Functionally, WT progenitor activity declined significantly from 6 to 12 months, and again at 2 years, with a threefold reduction in Matrigel colony-forming capacity."

Please note that our study used a mouse strain with similar genetic background and age group to the one referred to here - they used 9 week, 6 month, 1 year and 2 years old C57BL/6 mice. We used 9-week and 2 year old C57BL/6 mice. Therefore, our results are directly comparable to those used in the cited NCB paper.

"A subset of clock-controlled genes included those linked to progenitor/epithelial cell function, e.g. $\alpha 6$ -integrin, PrKC ϵ , P21, or Bcar3" needs references.

We added references to $\alpha 6$ -integrin, PrKC ϵ , P21, & Bcar3

Figure 4B, D and S9, instead of showing representative traces, the authors should use error bars on their plot. The authors should only show normalized data graphs. The Fig 5F should be also normalized.

We added error bars to the graphs in Fig 4B, 4D, S9; We normalised Fig 5.

REVIEWERS' COMMENTS:

Reviewer #1 (Remarks to the Author):

I accept the argument of the authors for the inclusion of the transcriptomic data in Figure 1, these results do have the potential to be a valuable resource for the field. I also agree with their point regarding the inclusion of data from Figure 2. The data from experiments using active and negative RhoA, or RhoA knockdown satisfactorily address my comment. In addition, the stem cell cultivation experiments in the presence or absence of ROCK inhibitor are satisfactory. With regard to the minor comments, I still think it would be useful to define CT in the manuscript text, so that when readers are looking at Figure 1C, it's immediately obvious. I also think it would be useful to put a label and ticks on the X-axis on the plot in Figure S1B. Otherwise, why not put these data in a table, which it essentially is. Neither of these minor points is critical, just suggestions.

However, I do think that the manuscript could be substantially re-organized to bring much of the data from the supplemental figures to the primary figures. I understand that it was originally prepared for a journal with very tight space limitations. But for Nature Communications, this is no longer an issue. It would make sense to bring as much of the important data forward and obvious.

Reviewer #2 (Remarks to the Author):

The authors of manuscript NCOMMS-16-11898A propose that some circadian clock genes are partly regulated by mechanical properties of the microenvironment. To do so they have employed a number of in vivo and in vitro experimental modalities. As one of the original reviewers of this work, I am much more convinced by the present manuscript, and I think the authors have provided some additional pieces of data that make their case more believable. Addition of the AFM studies and the comparison of 2 3D environments with 2D environments coated with multiple ECM are particularly helpful. It seems like it is the right time to give the rest of the scientific community access to these data so we can start to discuss them more broadly.

Additional comments:

I was surprised by the finding that the periductal region was stiffening with age, but the presence of more aligned collagen fibers is consistent with those measurements. In humans it is unclear whether stiffness changes with age in breast, but overall mammographic density tends to decrease with age. Are there any studies of these clocks in human breast cells as a function of age?

In the discussion the authors propose the YAP/TAZ or MRTF/SRF transcription factors may play a role in regulation/deregulation of the clock programs because they are known to play a role in mechanoresponse. YAP/TAZ are mainly in the basal/myoepithelial cells of mammary epithelia in mice and women, so does that mean the luminal cell clocks would be regulated differently? Is there any evidence that either sets of transcription factors are regulated differently with age?

Even in spite of the new data added it is pretty clear that microenvironment components other than mechanical forces are regulating the clock. Moreover, mice DO NOT have breasts. I would suggest a modification to the title such as: "Cellular mechano-environment is a regulator the mammary circadian clock"

REVIEWERS' COMMENTS:

Reviewer #1 (Remarks to the Author):

I accept the argument of the authors for the inclusion of the transcriptomic data in Figure 1, these results do have the potential to be a valuable resource for the field. I also agree with their point regarding the inclusion of data from Figure 2. The data from experiments using active and negative RhoA, or RhoA knockdown satisfactorily address my comment. In addition, the stem cell cultivation experiments in the presence or absence of ROCK inhibitor are satisfactory.

Many thanks for the positive comments.

With regard to the minor comments, I still think it would be useful to define CT in the manuscript text, so that when readers are looking at Figure 1C, it's immediately obvious.

As requested, we have defined CT in the M+Ms.

I also think it would be useful to put a label and ticks on the X-axis on the plot in Figure S1B. Otherwise, why not put these data in a table, which it essentially is. Neither of these minor points is critical, just suggestions.

Thanks for the suggestion. We've put the data into a table.

However, I do think that the manuscript could be substantially re-organized to bring much of the data from the supplemental figures to the primary figures. I understand that it was originally prepared for a journal with very tight space limitations. But for Nature Communications, this is no longer an issue. It would make sense to bring as much of the important data forward and obvious.

Yes, we've put some of the more fundamental figures that were in Supplementary Figures into the Main Figures.

We now have 10 Main Figures, and hope that this is OK.

Reviewer #2 (Remarks to the Author):

The authors of manuscript NCOMMS-16-11898A propose that some circadian clock genes are partly regulated by mechanical properties of the microenvironment. To do so they have employed a number of in vivo and in vitro experimental modalities. As one of the original reviewers of this work, I am much more convinced by the present manuscript, and I think the authors have provided some additional pieces of data that make their case more believable. Addition of the AFM studies and the comparison of 2 3D environments with 2D environments coated with multiple ECM are particularly helpful. It seems like it is the right time to give the rest of the scientific community access to these data so we can start to discuss them more broadly.

Many thanks for the positive comments.

Additional comments:

I was surprised by the finding that the periductal region was stiffening with age, but the presence of more aligned collagen fibers is consistent with those measurements. In humans it is unclear whether stiffness changes with age in breast, but overall mammographic density tends to decrease with age. Are there any studies of these clocks in human breast cells as a function of age?

There are no studies yet, that have been published on clocks vs age in human breast. Hopefully, we'll be able to do that in the near future.

In the discussion the authors propose the YAP/TAZ or MRTF/SRF transcription factors may play a role in regulation/deregulation of the clock programs because they are known to play a role in mechanoresponse. YAP/TAZ are mainly in the basal/myoepithelial cells of mammary epithelia in mice and women, so does that mean the luminal cell clocks would be regulated differently?

We've found that YAP is present in the luminal cells of both mouse and human mammary gland. Therefore we do not think clocks in luminal cells will be regulated differently. We are following this up with shRNA knockdown and phenotyping studies, which will be the subject of a future publication.

Is there any evidence that either sets of transcription factors are regulated differently with age?

Indeed, the results from *Pelissier et al. Cell Rep. 2014* support the hypothesis that the YAP pathway is regulated differently in ageing. Also we aim to look at this in human breast tissue in future studies.

Even in spite of the new data added it is pretty clear that microenvironment components other than mechanical forces are regulating the clock. Moreover, mice DO NOT have breasts. I would suggest a modification to the title such as: "Cellular mechano-environment is a regulator the mammary circadian clock"

As requested, we have changed the title to "Cellular mechano-environment regulates the mammary circadian clock".